# Affectation of COVID-19 pandemic on the use and abundance of wild resources in Tabasco, Mexico: A qualitative assessment

**José Luis Martínez-Sánchez**⬡*, **Carolina Zequeira Larios**⬡, **Florisel Hernandez Ramirez**⬡

División Académica de Ciencias Biológicas, Universidad Juárez Autónoma de Tabasco, Villahermosa, Tabasco, México

⬡ These authors contributed equally to this work.

* jose.martinez@ujat.mx

**Data Availability Statement:** All relevant data are within the manuscript and its Supporting information files (S1 Data).

## Abstract

Southern Mexico is particularly rich in natural resources, yet unemployment has risen to 8% during the COVID-19 pandemic. The effect of the pandemic on the use and abundance of Tabasco's wild resources was examined through personal surveys. By using Microsoft Forms® with cell phones 1,963 surveys were collected. Cronbach's alpha, Z-value, and chi$^2$ were calculated using the MAXQDA Analytics Pro program. A higher abundance of wild resources before the pandemic than today (57% vs. 11%) was observed. During the pandemic, people referred more to a high use (28%) of resources than to a low use (20%). This caused the low abundance or scarcity of wild products to be greater during the pandemic than before the pandemic (43% vs. 4%). Wild foods and timber were the most used products. The pandemic has produced a greater use of natural resources probably due to the high unemployment rate in rural areas. Future studies of wild products should address the relevant products in the locality and their even sampling. Finding suitable respondents is highly recommended.

## Introduction

### The COVID-19 in the world

It is common knowledge that at the end of 2019, a pandemic caused by the COVID-19 (hereafter refer either as COVID-19 or pandemic) virus originated in China, has been present in around 210 countries [1], with roughly (last update 10 march 2023) 676.6 million people having been infected and causing 6.9 million fatalities worldwide [2]. SARS-CoV-2 is the main pandemic threatening worldwide health [3]. COVID-19 presented many notable variants for their impact on human health, including the alpha, beta, delta, and omicron variants. This gave rise throughout the world to the development of many medical studies to discover the cause of the pandemic ([4, 5] as well as the implementation of protection and preventive measures [6]. A total of 13,338.8 million vaccines have been applied worldwide. According to the World Health Organization the pandemic has officially terminated as of May 2023.

**Funding:** This work was supported by the Council of Science and Technology of the State of Tabasco, Mexico [Consejo de Ciencia y Tecnologia del estado de Tabasco´], grant number: PRODECTI-2020-01/019.

**Competing interests:** The authors have declared that no competing interests exist.

### The COVID-19 in Mexico and Tabasco

In Mexico, figures report 7.5 million infections and 333,188 deaths [2], while the state of Tabasco has reported 191,977 infections and 6,244 deaths [7]. In the case of Mexico, the preventive measures for contagion have already been relaxed because as of May 14 2022, the use of a mask in public places is no longer officially mandatory [8]. Face masks and distancing have been reported as effective prevention measures [9–11]. As of today (last update 20 October 2021) 74% of Mexico's population over 14-yr old have been vaccinated.

### The effect of the pandemic on natural resources

COVID-19 has caused an economic recession, unemployment, lifestyle changes, and many other damaging outcomes at the global level. The condition before and impact after the COVID-19 pandemic have been studied on several topics like air quality, air pollution, environmental variables [12–14], but none on wild resources. The reduction of human activity and productive chains caused an increase in the price of natural resources [15]. Contrastingly, natural resources have thrived throughout the pandemic [16]. In particular, a trend towards increasing abundance of several animal species has been observed [17]. Human activities have pervasive effects on the diversity, abundance, and distribution of animals across landscapes [17–19]. Concerning birds, for example, [20] found an increase in the mean daily number of urban observations of birds during the 2020 pandemic lockdown in Italy and Spain, compared to previous years, which also accounted for more people contributing bird sightings while in lockdown [21].

The factors contributing to this increase in observations include decreased pollution, noise, activity, and human presence [16]. Due to this change in human activities, the pandemic has also directly impacted the use of natural resources [22]. Some examples are decreased forestry management activities and the exploitation of marine resources. Fishing dropped by 34%, landings by 49%, and revenue declined by 39% compared to the same period in 2017–2019 in the Mediterranean Sea of Spain during the lockdown [23]. Also, slight short-term biomass increases for fast-growing, small-sized organisms during 2020 were projected, reflecting the positive effects of reduced fishing with the relaxation of the lockdown [23]. Recently, [24] have found that indigenous peoples and local communities worldwide could maintain a sustainable use of their natural resources during the pandemic and achieve and guarantee their subsistence without significant problems.

### Natural resources in the state of Tabasco

Tabasco has an area of 25,267 km$^2$ and is located in southeastern Mexico with a tropical climate, average annual temperature, and rainfall of 27° C and 2,550 mm respectively. Fifty-six percent of the surface is grassland, 15.5% agriculture, 20.5% wetlands, 5% wooded vegetation, and the remaining 3% contain other types of vegetation, aquatic, and urban areas [25]. Therefore, abundant natural resources are widely and historically significant in the local lifestyle and economy. There is a large number of flora and fauna species in the state's various terrestrial and aquatic ecosystems, which are used as a natural resource by the inhabitants, of which to date there is no complete inventory. To date few works document the use and exploitation of Tabasco's natural resources. However it is generally known that these consist of timber, non-timber products, and terrestrial and aquatic flora and fauna, which are used for food, medicinal plants, and elaboration of handicrafts (S1 Appendix). For example, in the area called the Centla swamps, around 260 plant species have been identified, of which 76 are food or medicinal, and some can be used for construction, live tree fences and as fuel [26]. Recently [27], a mixture of three extracts of the plants *Psidium guajava*, *Camellia sinensis*, and *Rosa hybrida*

was recognized as a health food ingredient that alleviates COVID-19 symptoms. The use of medicinal plants, particularly in this pandemic, has been present [28].

## Socioeconomic condition of Tabasco during the pandemic

In the global context, "Many workers lost their jobs during the COVID-19 crisis, as their employers couldn't pay their salaries" [1]. One year into the current pandemic, as of March 2020, unemployment in Tabasco has increased to 8.02% [29], particularly in rural areas with the highest presence of natural resources.

## The use of surveys in the study of natural resources

Qualitative studies may be considered subjective, difficult to understand, or prone to biases [30, 31]; however, they remain present in scientific research to address human emotions in ways that quantitative studies cannot [32–34]. From 1991–2003, 127 articles were published in 22 scientific ecology journals based on surveys of plant and animal species and their conservation. The mean (± SE) sample size (number of respondents) per questionnaire was 1422 ± 261 [35]. This methodology has demonstrated that quantifying public perception is vital for translating ecology into management [35].

The present study used local knowledge; however, the validation or representativeness of qualitative estimates made by local communities has only been demonstrated in a limited capacity. Local knowledge on resource abundance, bound by place and time, retains its meaning when employed in a scientific context. This knowledge is relevant for decisions about natural resource management [36]. In an attempt to validate the use of qualitative surveys instead of quantitative sampling, [36] conducted a study of estimates of the abundance of natural resources of flora and fauna through direct sampling on a four-category scale (many individuals, ≥10, some individuals, 1–9, few individuals, 1, very few or no individuals) along 2 km transects. In two locations, this study was conducted by two Indigenous communities (qualitative) and a scientific researcher (quantitative). They found that the results were comparable and led to similar values and conclusions and that the Indigenous community's estimates were eight times less than the cost of the scientific estimate.

Other researchers have also used the term "local knowledge" when referring to recent knowledge and "Indigenous knowledge" to refer to the local knowledge of Indigenous peoples, or local knowledge unique to a culture or society [36, 37]. In studies concerning the human impacts on wild species and human behavior about wild species (i.e., perceptions towards the use and exploitation of natural resources), questionnaires often provide the best means for obtaining quantitative data from a large number of sites [35, 38]. According to [31, 39], qualitative research can better understand a complex phenomenon than empirically measuring it. [21] conducted 133 online global surveys in English, Spanish and French in 40 countries to see how COVID-19 affected indigenous peoples and local communities accessing and using natural resources and traditional medicine during the pandemic.

## The use of Microsoft Forms® and mobile phones in the application of surveys

Social media platforms such as YouTube, Facebook, and Twitter have attracted attention among researchers in many disciplines for conducting studies based on qualitative and quantitative methods (mixed methods) [40]. Recently [24], conducted a study on indigenous peoples and local communities applying the SenseMaker® software upon 133 surveys to see how COVID-19 affected the access and use of natural resources during the pandemic. This software enables the analysis of micronarratives based on how respondents classify their stories.

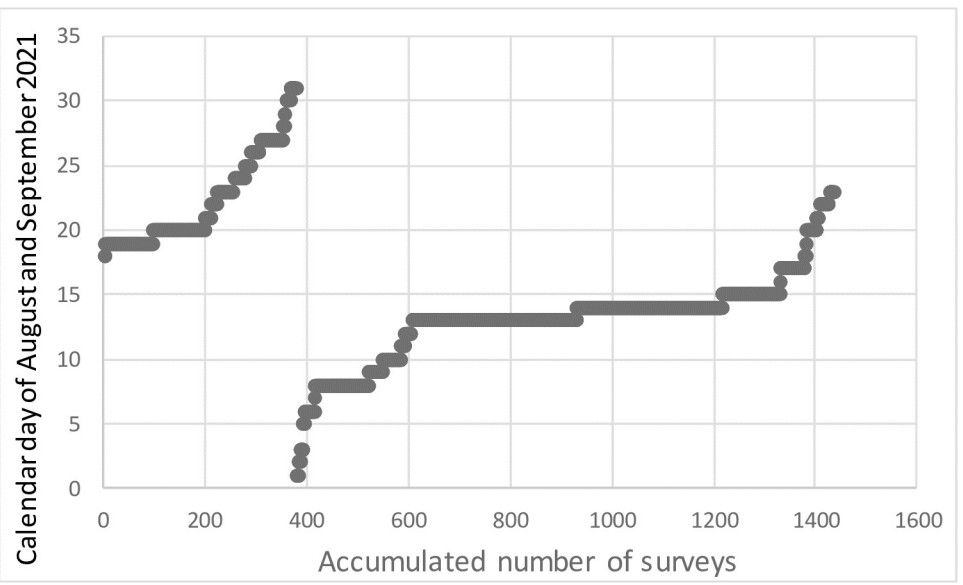

**Fig 1. Cumulative number of surveys completed from August 18 to September 23, 2021.**

In this study, Microsoft Forms® was another tool for conducting high-impact qualitative research. In just one calendar day, it allowed to collect over 350 surveys (Fig 1). Microsoft Forms® is a low-cost, highly feasible, and short-term implemented approach for qualitative research interests and possesses great potential for understanding phenomena like the effects of the pandemic. It allows researchers direct access to people *in situ* and in real-time. The method is suitable for qualitative research projects of considerable size. We have demonstrated that using Microsoft Forms® on mobile phones can be a powerful digital tool for employing classic qualitative methods. Additionally, it provides a flexible and discrete interview mode as a mobile method. Most approaches for qualitative methods using mobile devices stem from quantitative origins [41–45]. The increasing use of the internet has prompted methodological innovations in data collection, which can now be undertaken at any time and at low cost, for qualitative research on platforms such as Microsoft Forms®, where data are available immediately, and require no transcription for participants who are geographically dispersed [46, 47]. Online surveys are recognized as a method that provides rich data, comparable to traditional surveys, and anonymity, which helps to reduce participant inhibitions [47–50]. [51] states that face-to-face interviews represent the best qualitative methodology for conducting interviews, while other modes constitute the 'second best' [51, 52].

## The purpose of this study

This study aims to determine the impact of the current pandemic on the use and abundance of local natural resources in Tabasco, through possible changes in extraction habits before and during the pandemic. The study is not based on directly estimating of resource abundance, but on subjective appreciation through surveys of a representative population of users. In the present study, the subject of natural resources is analyzed under the scientific concept of cause and effect, where the use of natural resources is the cause of their relative abundance (effect). Until today there is no study that has compared the condition of natural resources before and after a pandemic, so this study would be the first of its type. [53] study on sustainable rural community-based ecotourism in Sabah, Borneo, is the closest study to quote during the pandemic.

## Objective

To compare the use and abundance of various categories of natural resources, before and during the COVID-19 pandemic in Tabasco, Mexico.

Hypothesis 1: During the current pandemic people stayed at home for fear of the virus, and the economy and sales of products from natural resources decreased, and consequently decreased their exploitation and use, which could lead to a recovery in the abundance of natural resources (i.e., less use can cause greater abundance of natural resources).

Hypothesis 2: During the current pandemic, the local unemployed population increased, and consequently, the use of natural resources for obtaining food and selling products increased, reducing the abundance of natural resources (i.e., more use can cause less abundance of natural resources).

## Materials and methods

The following qualitative evaluation was carried out with the objective to estimate the impact of the COVID-19 pandemic on the use and abundance of wild resources in Tabasco, Mexico.

Questionnaires, or social surveys, are used in ecology to test research hypotheses when information is required from a specific human target population, in this case, users of natural resources. Thus surveys are suitable for approaching public or stakeholder perceptions in ecological management, studies of human impacts on wild species, and interdisciplinary studies that include ecological and non-ecological components [35]. Surveys provide vital information for all kinds of public information and research fields [49, 50, 54, 55] and particularly for COVID-19 [56]. According to [51, 57], inquiring in qualitative research relies on Interviewing.

A 27-question survey (S1 File, link survey) was prepared on the use and abundance of natural resources before and during the pandemic. This study defines a natural resource as any wild product extracted from the environment without prior management or cultivation. The survey format was based on [58, 59]. The survey was designed in Microsoft® Forms and distributed to users of natural resources. Responses were collected anonymously. Anonymity reduces inhibitions [60, 61] and increases the confidence that subjects' responses will remain private [61, 62]. Data were collected through mobile devices. Four questions covered respondents' demographic information (locality, gender, age), 16 questions addressed the degree of effect the pandemic had on the abundance and use of five categories of natural resources, seven questions focused on sanitary measures and habits adopted in the face of the pandemic and one final open question. Six questions and their responses were dichotomous (yes, no; male, female), seven open (four text and three numeric), and 14 were ordinal on a Likert scale with three and four options (high, moderate, low; less, the same, more; always, sometimes, never; weekly, fortnight, monthly, year) [63]. A Likert scale is an ordered scale from which respondents choose one option that best aligns with their view [55, 64]. We used the Cronbach's alpha test [65] in MAXQDA Analytics Pro 2020 (Release 20.4.1) to validate the scales' reliability. Specifically we compared the variances of the responses (Eq 1) [66].

$$\alpha = K/K - 1(1 - \Sigma Vi/Vt) \tag{1}$$

where K is the number of items or questions, Vi is the variance of each item and Vt is the total variance of the scale.

The index takes values between 0 and 1, the ideal range between 0.8 and 0.9, since below 0.8 offers low representativeness, and above 0.9 represents redundancy in the survey questions [65, 67].

We followed the recommendations of [35]: (i) the definition of the target population (users of natural resources) and any hypotheses to be tested should be documented; (ii) questionnaires should be piloted before their use; (iii) the sample size should be sufficient for the statistical analysis; (iv) the number of non-respondents should be minimized; (v) the question and answer format should be kept as simple as possible; (vi) the structure of the questionnaire and the data emerging from it should be unambiguously shown in any publication (link). Field surveys can provide information on current impacts or management practices, but are usually unsuitable for revealing information from the past. Questionnaires can be a vital research technique [35].

The pandemic's start date was March 20, 2020, although its official start date in Mexico was March 27, 2020 [68]. The survey was provided through a link sent by email to all undergraduate students over 18-yr old from 10 higher education schools (half of the total schools in the state) from 10 municipalities of Tabasco. The selection of these universities was based on them being distributed and covering most of the state. In this way, the users of natural resources surveyed by the students of these universities finally came from the 17 municipalities that make up the entire state of Tabasco. The above could be achieved because some users from other municipalities were in transit and could be surveyed. The students then took the survey home to apply it to whoever adults over 18 used or exploited natural resources in the region. The survey was conducted over 46 days (August 18 to October 7, 2021) and graphed until September (Fig 1). In addition, public markets in three state municipalities were visited, where sellers of wild natural products over 18-yr old were also surveyed.

Ethics statement.- The present study is the results of the publication of the respective research project [*PRODECTI-2020-01/019*] approved, reviewed and financed by an institutional review board named the Science and Technology Council of Tabasco, Mexico.

## Data analysis

The data (categorical type) were analyzed based on methods described by [63]. To compare two proportions or percentages and to obtain the confidence interval of their difference, the standard error (SE, Eq 2) of their difference and the critical value of Z (1.96) were calculated at the value of P = 0.05 [69]. Any value of Z > 1.96 is significant at P < 0.05. The chi$^2$ test was also used [70].

$$SE(P_1 - P_2) = \sqrt{\frac{P_1(100 - P_1)}{n_1} + \frac{P_2(100 - P_2)}{n_2}} \qquad (2)$$

Z = difference between percentage values/SE (Eq 2)

**Ethical disclosure.** Here we declare that no formal or written consent was necessary from the participants because they only responded to a 7-minutes friendly face to face survey with a short talk. Before the application of the study, only verbal consent was requested. All the participants were over 18-yr old.

## Results

### General data

A total of 1,963 surveys were completed from 1,863 villages within the 17 municipalities of the state of Tabasco. The questionnaire had an average completion time of 7 min 45 sec. Fifty-four percent of natural resource users were men and 46% were women. The users were between 17 and 91 years old with a mode of 21 years old, and 50.5% of respondents were between the ages

of 17 and 22. From the initiation of the surveys during the pandemic, 59 new users of natural resources who had yet carried out to this activity were registered.

The 14 questions using a 3 or 4 option Likert scale presented a Cronbach's α index of 0.89 and 0.85, respectively.

The natural products with greater frequency of use (weekly) before and during the pandemic were wild foods and wood products. Twenty-two percent of those surveyed used various fruits or firewood throughout the year, while 78% used all kinds of resources during specific months. The months with the highest use were March, April, and May (14% each), and the lowest use was observed in November and December (3% each). Sixty-five percent of those surveyed use natural resources for family consumption, 20% use resources as part of their economic income, and only in 15% of respondents did use constitute the primary family economic income. Natural resources are destined mainly for food consumption (45%), and various uses (45%), and only 10% are destined for sale in some way, of which half reported a decrease in sales due to the pandemic, and only 2% reported an increase.

Eighty-one percent of those surveyed indicated that they increased their time spent at home during the pandemic compared to before. At the same time, only 6% stayed less time at home during the pandemic (Z = 30.7, P = 0.000, IC = 70.2–79.8%), and 13% reported the same time (Fig 2). Ninety-two percent reported using sanitary measures whenever they left home, 7% sometimes, and only 1% reported never using them. Thirty-four percent reported to have contact with more than five people. Forty-three percent of the respondents reported having no family member with COVID-19, while 57% reported at least one member; the average was 1.71 members per family with a mode of 2 members and several families had > 10 infected members. About six percent (6.1%) of those infected declared the death of at least one family member at home due to COVID-19.

A relationship was present between respondents having contact with > 5 people and the presence of COVID-19 at home ($\chi^2$ = 11.9, P = 0.001, d.f. = 1). However this cannot be established as a cause-effect.

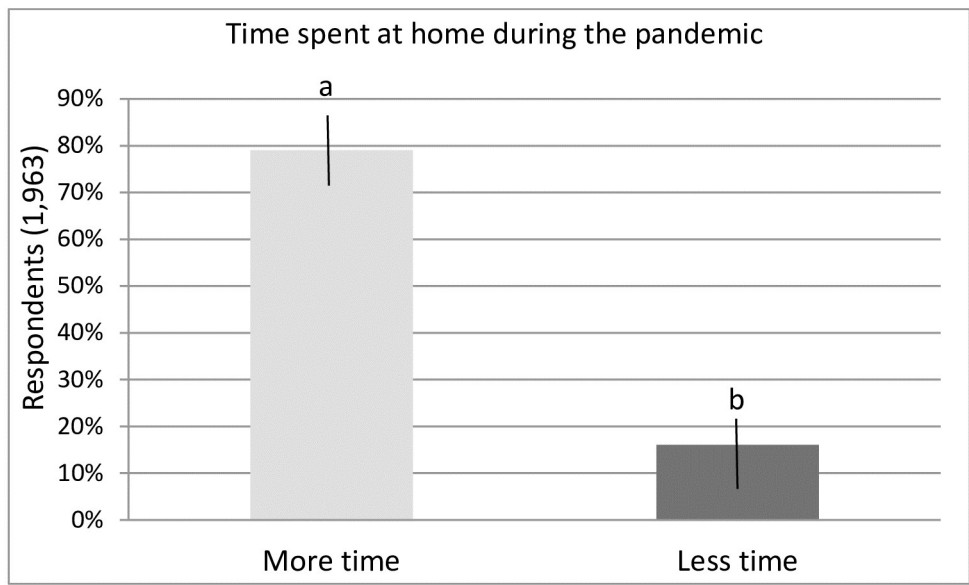

**Fig 2. Percentage of respondents who spent more and less time at home during COVID-19 compared to before the pandemic.** Different letters indicate the difference from the highest value using the Z test at P < 0.05.

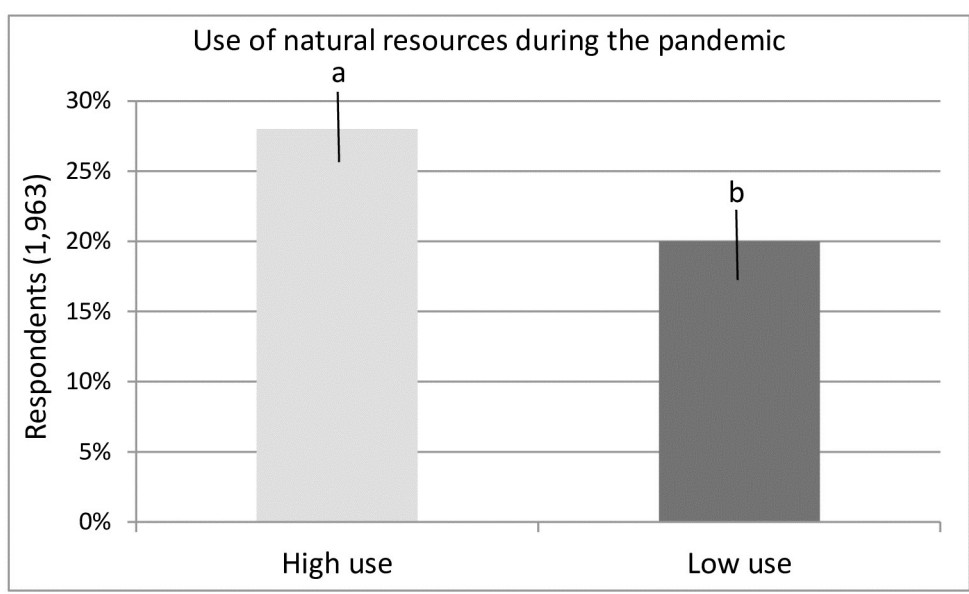

**Fig 3. Percentage of respondents who had a qualitative appreciation of greater or lesser use of natural resources during the COVID-19 in Tabasco, from March 20, 2020.** Different letters indicate the difference from the highest value using the Z test at P < 0.05.

## Use and abundance of natural resources before and during the pandemic

During the pandemic, the appreciation in the use of natural resources was more of a high use than a low use (28% vs. 20%, Z = 2.39, P = 0.01), with a difference of 8% (CI = 1.4–14.5%) (Fig 3).

The appreciation of the abundance of natural resources was higher before the pandemic than it is today (57% vs. 11%, Z = 16.9, P = 0.000), with a difference of 46% (CI = 40.6–51.3%), while the appreciation of low abundance or scarcity of resources is greater during the current pandemic than before the pandemic (43% vs. 4%, Z = 12.0, P = 0.000), with a difference 39% (CI = 34.7–43.7%) (Fig 4).

## Use and abundance of natural resources by type of product, before and during the pandemic

The percentages of the use and abundance of natural resources were similar for the different categories of products (Table 1). The response rate for high utilization before the pandemic ranged from 28% to 35% depending on the product category, and from 27% to 29% during the pandemic. For low use, the percentage ranged from 11% to 15% of the respondents before the pandemic and from 17% to 19% during the pandemic. For high abundance, it ranged from 55% to 57% before the pandemic and from 12% to 13% during the pandemic, and for low abundance, from 4% to 5% before the pandemic and from 28% to 33% during the pandemic (Table 1).

The differences in the use of resources before and during the pandemic were only observed in two categories of natural resources. In contrast, the differences in abundance were seen in all categories at the high and low levels. Before the pandemic, the high abundance of natural resources predominated and was more significant than during the pandemic, while during the pandemic the category of low abundance of natural resources was more common (Table 1).

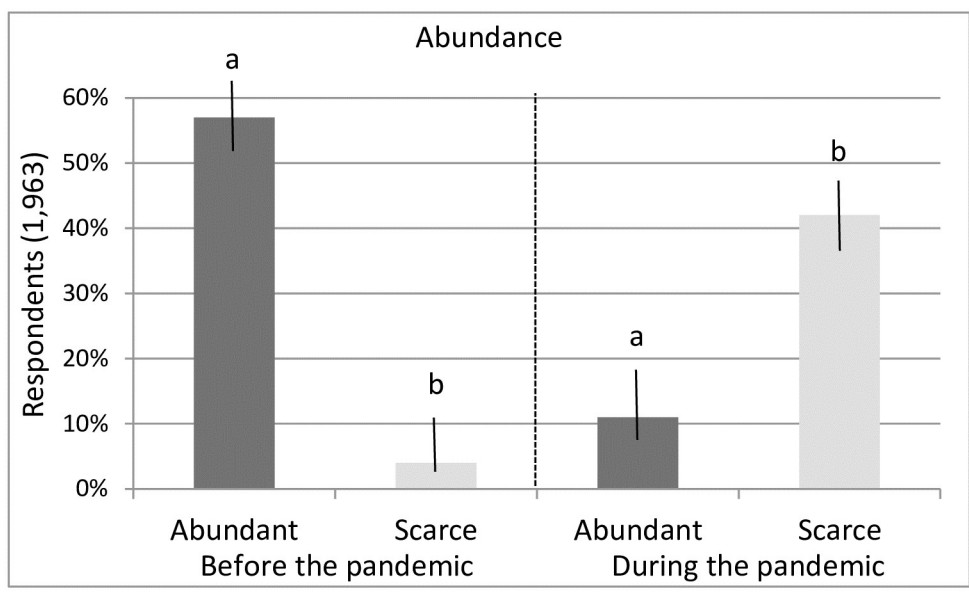

**Fig 4. Percentage of respondents who had a qualitative appreciation of the abundance (abundant or scarce) of natural resources during the pandemic compared to before the pandemic in Tabasco, Mexico.** Different letters indicate the difference from the highest value using the Z test at P < 0.05.

**Table 1. Percentages of the respondents (1,963) who had a qualitative appreciation of high or low use and abundance of the different natural resources before and during the pandemic in the state of Tabasco.**

| Use of natural resources | | Medicinal plants | Timber products, fire wood, coal | Edible wild animals | Plants, flowers, seeds, palm leaves | Edible wild plants and fruits | Other products |
|---|---|---|---|---|---|---|---|
| High | Before | 34% | 35% | 32% | 34% | 35% | 28% |
| | During | 29% | 27% | 29% | 29% | 28% | 27% |
| | Z | 1.66 | **2.7** | 0.89 | 1.61 | **2.4** | 0.28 |
| | P | 0.07 | 0.01 | 0.11 | 0.07 | 0.01 | 0.4 |
| Low | Before | 13% | 12% | 12% | 12% | 11% | 15% |
| | During | 18% | 17% | 18% | 17% | 17% | 19% |
| | Z | 1.49 | 1.53 | 1.61 | 1.42 | 1.85 | 0.82 |
| | P | 0.07 | 0.07 | 0.07 | 0.08 | 0.06 | 0.11 |
| Abundance | | | | | | | |
| High | Before | 57% | 56% | 55% | 57% | 57% | 57% |
| | During | 13% | 12% | 12% | 13% | 13% | 13% |
| | Z | **14.9** | **15.2** | **13.1** | **14.6** | **15.4** | **10.0** |
| | P | 0.000 | 0.000 | 0.000 | 0.000 | 0.000 | 0.000 |
| Low | Before | 5% | 5% | 4% | 5% | 5% | 4% |
| | During | 32% | 33% | 32% | 31% | 32% | 28% |
| | Z | **12.5** | **8.6** | **7.9** | **7.6** | **8.4** | **4.8** |
| | P | 0.000 | 0.000 | 0.000 | 0.000 | 0.000 | 0.008 |

Before and after values were compared with Z > 1.96 and P < 0.05 and their significant differences are shown in bold.

Wild foods and timber products were the most frequently use (weekly) natural products before and during the pandemic.

## Discussion

### Survey sample size

Considering that there are numerous significant qualitative studies with a relatively small sample size (< 30 surveys) i.e. [33, 34, 58, 71, 72], the present study (1,963 direct surveys) constitutes a large qualitative data set (> 100 surveys *sensu* [73]). Studies based on 485 surveys with 17 closed-ended questions with two to four qualitative options have provided significant results [74]. However, it is essential to mention that in this study a disproportion was obtained in the number of surveys completed in the different geographical areas due to the different population densities. To reduce this bias in future studies; it is recommended to use an equal number of surveys for all geographic areas.

The present study allowed qualitative estimation of the abundance of natural resources in the state of Tabasco, which would be practically impossible to conduct objectively or quantitatively since it would require costly and time-consuming sampling to acquire direct data on the extensive different terrestrial and aquatic ecosystems and natural resources in the region. For this reason, [36] conducted a qualitative study to replace a quantitative estimate, obtaining significantly similar results. Likewise, in the present study, the direct estimation of the use of natural resources would require sampling in multiple stores and selling points throughout the state.

Thus, given the reasoning, the present study of almost 2,000 direct surveys is considered methodologically valid.

### Use and abundance of natural resources during the pandemic

The present study, unlike other localities reported [16, 20, 23] showed that the current pandemic did cause damage to the natural resources of the state of Tabasco, as these resources are now estimated to be less abundant compared to before the pandemic. Before the pandemic, a high abundance of natural resources dominated. During the pandemic, low abundance dominated (Fig 3). Supporting our finding, only [75] recently reported that the COVID-19 outbreak has had considerable negative impacts on the livelihoods and living conditions of communities around the world, particularly on the wild meat in Cameroon. The population surveyed in our study represented 0.2% of the economically active population (18+ years) of the state of Tabasco (967,637 people) as of 2021 [76], and corresponds to the population dedicated to the use and exploitation of natural resources in the state.

Hypothesis 2 was confirmed, suggesting that the loss of employment during COVID-19 and an increase in the population of local users of natural resources for food and natural products to sell, could cause a decrease in the abundance of natural resources. During the pandemic, respondents referred to a scarcity of natural products more than their pre-pandemic abundance (Fig 3). The condition of natural resources changed from high abundance before the pandemic to moderate abundance during the pandemic, as the high abundance condition decreased from 58% to 13% of respondents. The pandemic increased the low abundance of natural resources from 4% to 31%. The appreciation of high abundance of natural resources decreased with the pandemic in 46% of those surveyed.

Not all types of products demonstrated the same pattern. In some products, use decreased while in others increased, and in some products, abundance decreased and in others increased. The changes in the abundance of the different types of products (Table 1) influenced their general pattern of abundance (Fig 3).

It has been argued that the current pandemic arose from the consumption of bush-meat in China, so there is an invitation to restore and strengthen the natural protected areas up to 50% of the world's land area [77], which would result in the conservation of biodiversity, reduction of climate change, improvement of human health at a global scale, and the reduction of human contact with zoonotic pathogens. A study of 40 Ebola outbreaks since 2004 found that they were significantly linked to the recent clearing of mature forest, leading to more frequent contact between humans and infected animals [77, 78]. [77] proposals would enable humans to live in a healthy balance with nature leading to a long-term resilient future.

With the information quoted above, a call is made to improve the sanitary measures in the movement of living animal and vegetable beings for food use, from their collection in the field through the selling points in public markets, as well as in its possible re-introduction or movement in the reverse direction. It has been proven that the flow of pathogens can be in both directions [1].

## Socioeconomic condition of Tabasco during the pandemic

Tabasco stands out as one of the 31 states in Mexico with a large percentage area of the state (711,675 ha; 28%) [79] vs. urban area (< 1%), where there is still a large abundance of natural resources, and where local people use and live much from these resources. Likewise, it stands out as one of the states of Mexico that have suffered the most economic damage and job loss with the current pandemic since during the fourth quarter of 2020; Tabasco ranked as the second region with the highest rate of unemployment (8%) of the country [80].

These two factors, the high availability of natural resources near human settlements and the high loss of employment during this pandemic, present possible causes for the damage to the abundance of natural resources, in contrast to the recovery documented in other locations. Among the data that can help understand this phenomenon is that in Tabasco there is a relatively higher proportion of rural communities, with 2,324 rural localities *vs*. 148 urban; 59% of the population of the state of Tabasco live in urban areas vs. 41% in rural, compared to the national total of 79% living in urban localities *vs*. 21% in rural areas [76].

## Strengths and limitations

The present study shows the possibility of obtaining rapid and realistic ecological information from a large rural area with places that are geographically difficult to access, through the use of mobile devices and informatics by many surveyors.

As a limitation of the study, it can be considered finding respondents who know the resources well and share their information to have better result and diagnoses of the topic.

## Future research directions

It is recommended that future studies emphasize sampling of the most essential products in the locality, and searching for respondents of the specific gender and age to obtain the best quality information. Also, applying a similar sample size to the different geographical areas of the study would be highly desirable.

## Conclusions

From this research, the hypothesis 2 about the adverse effect of unemployment on rural natural resources was confirmed, and it is concluded that human pandemics can cause damage to natural resources when populations are highly linked to the local socioeconomic factors and settled near a high wealth, as in the case of Tabasco, Mexico.

COVID-19 did affect the use and abundance of natural resources in a rural region of south-eastern Mexico, the most affected products being medicinal plants wild plants, and timber.

This study can be considered a current contribution to the socioeconomic effect of the pandemic on natural resources, since it is the first of its kind so far, having found an adverse impact of unemployment in rural areas.

## Supporting information

**S1 Appendix. Common names of the most common species in their use and exploitation in the state of Tabasco, México.**
(DOCX)

**S1 File. English version survey.**
(PDF)

**S1 Data.**
(XLSX)

## Author Contributions

**Conceptualization:** José Luis Martínez-Sánchez.

**Data curation:** Florisel Hernandez Ramirez.

**Formal analysis:** José Luis Martínez-Sánchez.

**Methodology:** Carolina Zequeira Larios.

**Project administration:** José Luis Martínez-Sánchez.

**Writing – original draft:** José Luis Martínez-Sánchez.

**Writing – review & editing:** José Luis Martínez-Sánchez.

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
