## [Decision Letter · Decision Letter 0]

26 Dec 2023

PONE-D-23-33512Socioeconomic effect during the Covid-19 pandemic on the use and abundance of natural resources in Tabasco, Mexico: A qualitative assessmentPLOS ONE

Dear Dr. Matinez,

Thank you for submitting your manuscript to PLOS ONE. After careful consideration, we feel that it has merit but does not fully meet PLOS ONE’s publication criteria as it currently stands. Therefore, we invite you to submit a revised version of the manuscript that addresses the points raised during the review process.

We look forward to receiving your revised manuscript.

Kind regards,

Bert B. Little, MA, PhD, FAAAS, FRAI, FRSM, FRSPH

Academic Editor

PLOS ONE

Journal Requirements:

"This work was conducted and supported by the Council of Science and Technology of the State of Tabasco, Mexico [Consejo de Ciencia y Tecnologia del estado de Tabasco, PRODECTI-2020-01/019]. "

“The authors received no specific funding for this work.”

5. We note that your Data Availability Statement is currently as follows: [All relevant data are within the manuscript and its Supporting Information files.]

Additional Editor Comments:

The expert review indicates a Major Revision.

Statistical methods and results need improvement.

I would suggest a reanalysis.

The writing is judged as also needing major revision.

Suggested references to provide better context.

Reviewers' comments:

Reviewer's Responses to Questions

**Comments to the Author**

1. Is the manuscript technically sound, and do the data support the conclusions?

Reviewer #1: Yes

Reviewer #2: Yes

2. Has the statistical analysis been performed appropriately and rigorously? 

Reviewer #1: Yes

Reviewer #2: No

3. Have the authors made all data underlying the findings in their manuscript fully available?

Reviewer #1: No

Reviewer #2: No

4. Is the manuscript presented in an intelligible fashion and written in standard English?

Reviewer #1: Yes

Reviewer #2: No

5. Review Comments to the Author

Reviewer #1: - Data availability statement should include a link to access the data in a public portal if no restrictions as stated

- Table 1, consider incorporating corresponding p-values

- Ethics: Consent of human participants is vital and needs to be mentioned

- Questionnaire and consent forms need to be appended as supplementary materials

Reviewer #2: - Your manuscript needs some revision.

- The title of this study should be modify.

- Abstract should clearly inform the important findings in the present study.

- The lengthy sentences may be split in to smaller sentence without change of its meaning.

- Your key words seem to be general and should be revised based on MESH.

- Background: The introduction section should be revised.

- Should add some new references published in PLOS ONE.

You can add the following references:

- Lak E, Mohammadi MJ, Yousefi H. Impact of COVID-19 acute respiratory disease on the risk factors attributed to cancer patients. Toxicology Reports. 2022 Jan 1;9:46-52.

- Yazdani M, Baboli Z, Maleki H, Birgani YT, Zahiri M, Chaharmahal SS, Goudarzi M, Mohammadi MJ, Alam K, Sorooshian A, Goudarzi G. Contrasting Iran’s air quality improvement during COVID-19 with other global cities. Journal of Environmental Health Science and Engineering. 2021 Dec;19(2):1801-6.

- Goudarzi G, Babaei AA, Mohammadi MJ, Hamid V, Maleki H. Geographical and meteorological evaluations of COVID-19 spread in Iran. Sustainability. 2022 Apr 30;14(9):5429.

- Neisi A, Goudarzi G, Mohammadi MJ, Tahmasebi Y, Rahim F, Baboli Z, Yazdani M, Sorooshian A, Attar SA, Angali KA, Alam K. Association of the corona virus (Covid-19) epidemic with environmental risk factors. Environmental Science and Pollution Research. 2023 May;30(21):60314-25.

- Rahimi Z, Mohammadi MJ, Araban M, Shirali GA, Cheraghian B. Socioeconomic correlates of face mask use among pedestrians during the COVID-19 pandemic: An ecological study. Frontiers in public health. 2022 Nov 16;10:921494.

- Shirali GA, Rahimi Z, Araban M, Mohammadi MJ, Cheraghian B. Social-distancing compliance among pedestrians in Ahvaz, South-West Iran during the Covid-19 pandemic. Asian Journal of Social Health and Behavior. 2021 Oct 1;4(4):131.

- Rahimi Z, Shirali GA, Araban M, Mohammadi MJ, Cheraghian B. Mask use among pedestrians during the Covid-19 pandemic in Southwest Iran: an observational study on 10,440 people. BMC Public Health. 2021 Dec;21:1-9.

- Abbasi-Kangevari M, Malekpour MR, Masinaei M, Moghaddam SS, Ghamari SH, Abbasi-Kangevari Z, Rezaei N, Rezaei N, Mokdad AH, Naghavi M, Larijani B. Effect of air pollution on disease burden, mortality, and life expectancy in North Africa and the Middle East: a systematic analysis for the Global Burden of Disease Study 2019. The Lancet Planetary Health. 2023 May 1;7(5):e358-69.

- Borsi SH, Goudarzi G, Sarizadeh G, Dastoorpoor M, Geravandi S, Shahriyari HA, Mohammadi ZA, Mohammadi MJ. Health Endpoint of Exposure to Criteria Air Pollutants in Ambient Air of on a Populated in Ahvaz City, Iran. Frontiers in Public Health. 2022;10.

- Nikmanesh Y, Mohammadi MJ, Yousefi H, Mansourimoghadam S, Taherian M. The effect of long-term exposure to toxic air pollutants on the increased risk of malignant brain tumors. Reviews on Environmental Health. 2022 Jun 28.

- Materials and Methods: The name of study should be brought in methods section.

- Materials and Methods: please add the time duration of study.

- Materials and Methods: Please describe how the location of sampling was selected, in details.

- Materials and Methods: Statistical analysis of sample data should be modified. This section is unclear.

- Result: The results section should be modified.

- Result: Carefully check that all Tables.

- Result: Please define the abbreviation.

- Discussion: The discussion part should modify.

- Discussion: Refer to more updated articles on similar studies in the discussion section and also reference list.

- Discussion: Please highlight your study's strengths and limitations.

- Discussion: Suggest adding a paragraph on directions for future research, practice and policy.

- Conclusions should be short with important observations.

6. PLOS authors have the option to publish the peer review history of their article (what does this mean?). If published, this will include your full peer review and any attached files.

Reviewer #1: No

Reviewer #2: No

---

## [Author Response · Author response to Decision Letter 0]

25 Jan 2024

MS was revised and corrected following the two style pdf templates. 

The Funding Information was amended, withdrawn from the MS and placed in the Cover Letter and online submission. 

As an important issue, the editor request to reanalyze the data and to improve the statistical methods. However throughout the review, this point is never specified or a statistical procedure suggested, and then I could not define a way to make it. In such case, I would kindly request to indicate which procedures to follow instead. 

Anyway, I would like to justify the statistical analysis used in the study. Since the results (Table 1, Figures 2 – 4) are percentages, which takes values within a range 0 – 100 (0 – 1.0) and do not constitute an infinitive variable, a non-parametric test like Z test is recommended instead of parametric analysis by means or medians comparison. Their Z standard errors and respective P values were obtained (pg. 13). For percentage data which came from a Likert scale (classes of abundance, impact and use) a Cronbach's alpha test was first applied to ensure the reliability of the scale classes performed. All this is commented within the Methods section (pg. 11).

The writing was revised and modified hoping is now better. 

Eight out of ten suggested references, plus others Plos One references, were incorporated to the MS. 

Response to Reviewers

Raw data was made available through a supplementary S3 file. 

P values were incorporated to Table 1, and title and footnote arranged according to journal style. 

Original questionnaire (link, pg. 11 of MS) and english version questionnaire (S2 file), and Consent letter were added as supplementary files. 

Reviewer 2:

As suggested by the reviewer, the title was modified. 

Abstract was improved as suggested.

Keywords were replaced using MESH tool.

Ten references were suggested to incorporate to the Background and eight references were included considering its pertinence. Plus this, two more recent references from Plos One were also included. 

The name of study was brought in methods section at the beginning. Sorry, this recommendation was not very clear for me. 

The reviewer asked to add the time duration of study. However this was already mentioned on line 291 of the MS.

The reviewer asked to please describe how the location of sampling was selected, in details. This was amended in page 14 of the revised MS. 

The reviewer stated “Statistical analysis of sample data should be modified”, but there is nothing else suggesting how to improve the statistical procedures. In this way, I unfortunately could not select a different procedure to reanalyze data based on the objectives of the study, which I personally consider that these can be supported with the current statistical methods. Specifically, the paragraph from line 242 – 244 was unclear. It was rewritten.

The unique table of the MS was improved. 

The reviewer stated “Refer to more updated articles on similar studies in the discussion section and also reference list.” Actually discussion section has about 10 references from 2020 to 2022, and I could only added one suitable reference from 2022 since this particular topic still remains scarcely studied and reported worldwide. 

Study's strengths and limitations were improved and included at the end of the MS. 

Future research directions were also improved and included at the end of the MS.

Conclusions were shortened keeping the relevant findings of the study.

---

## [Editor Report · Decision Letter 1]

16 Feb 2024

Affectation of COVID-19 pandemic on the use and abundance of wild resources in Tabasco, Mexico: A qualitative assessment

PONE-D-23-33512R1

Dear Dr. Matinez,

We’re pleased to inform you that your manuscript has been judged scientifically suitable for publication and will be formally accepted for publication once it meets all outstanding technical requirements.

Kind regards,

Bert B. Little, MA, PhD, FAAAS, FRAI, FRSM, FRSPH

Academic Editor

PLOS ONE
---

## [Editor Report · Acceptance letter]

28 Feb 2024

PONE-D-23-33512R1 

PLOS ONE

Dear Dr. Matinez, 

I'm pleased to inform you that your manuscript has been deemed suitable for publication in PLOS ONE. Congratulations! Your manuscript is now being handed over to our production team.

Kind regards, 

on behalf of

Professor Bert B. Little 

Academic Editor

PLOS ONE